# Diagnostic Imaging of Pregnant Women and Fetuses: Literature Review

**DOI:** 10.3390/bioengineering9060236

**Published:** 2022-05-28

**Authors:** Eunhye Kim, Brenda Boyd

**Affiliations:** 1Department of Health Safety Convergence Science, Korea University, Seoul 02481, Korea; eunhklim@korea.ac.kr; 2Department of Radiation Science, Loma Linda University, Loma Linda, CA 92350, USA

**Keywords:** diagnostic imaging, embryo, fetus, pregnant women, radiation risk, radiology, X-ray

## Abstract

Healthcare providers have acknowledged the dangers of radiation exposure to embryonic and fetal health, yet diagnostic imaging of pregnant women is increasing. Literature that pertains to the topic of interest was reviewed to collect tertiary data. The purpose of this literature review was to present the various radiation risks for pregnant women and the fetus depending on the gestational age of the pregnancy. The specific effects of radiation on pregnant women and the fetus, X-ray risks depending on the gestational age of the pregnancy, and other potential health effects when performing diagnostic imaging procedures on pregnant women were discussed in this review. In addition, ethical issues have been considered by improving overall communication to minimize unnecessary radiation exposure to pregnant women and fetuses.

## 1. Introduction

Thanks to technological improvements, overall radiation doses from medical procedures has been increased in the past 25 years [1]. The American Society of Radiologic Technologists reported in the Radiation Safety Compliance Journal that the overall collective dose has increased nearly sixfold since the early 1980s. Indeed, more than 90% of the radiation exposure from unnatural sources is due to medical imaging; the greatest contributor to the increase in ionizing radiation is computed tomography (CT) [1,2]. Since patients are exposed to radiation during diagnostic imaging, healthcare providers must fully understand the various benefits and risks of diagnostic imaging.

Diagnostic imaging provides crucial information about diseases. During pregnancy, a woman’s body continuously changes and can face various diseases [3,4]. Diagnosing disease and providing proper medication for pregnant women is important to their health and that of the fetus. In order to make better decisions, physicians, radiologists, and radiologic technologists must have a better understanding of the benefits and harmful effects that diagnostic radiation exposure can have on pregnant women. For instance, abdomen X-rays expose pregnant women to primary radiation, and chest X-rays expose their abdomen and fetus to scatter radiation [5]. Although radiation exposure may be small, the risks tend to increase with total exposure, repeated exposure, and younger patients [6,7].

Even though many healthcare providers are now aware of the dangers that diagnostic radiation exposure can have on fetuses and pregnant women, diagnostic imaging of pregnant women continues to rise. When physicians request a diagnostic radiology order on pregnant women and have not obtained their consent, radiologic technologists must explain to them the harmful effects that radiation can have on fetuses and pregnant women and, ultimately, obtain their consent.

The American Registry of Radiologic Technologists published a policy notifying its healthcare providers about the harmful effects of radiation on pregnant women and fetuses. It states that the maximum radiation dose for a fetus is 100 mSv (Sievert) [8]. The decisions of both the physician and radiologist are a crucial part of the diagnostic imaging process.

X-ray risks to the fetus vary depending on the gestational age of the pregnancy at the time of exposure. The highest risk for fetal abnormalities from radiation exposure occurs between 8–15 weeks of gestation [2,9,10,11]. The most harmful radiation effects are caused by the primary radiation beam on the abdomen of a pregnant woman. Fetal effects can also be caused by low-level radiation and can increase the risk of childhood cancer, as shown through animal and human studies [9,12,13,14,15,16,17,18,19,20,21]. Most healthcare providers acknowledge that scatter radiation exposure is not strong enough to affect an embryo or fetus. The double usage of shielding helps protect embryos and fetuses from scatter radiation, but a lower level of radiation could still affect them. Physicians and radiologists have to decide whether it is in a patient’s best interest to expose them to radiation by comparing the benefits and risks of medical X-ray imaging.

## 2. Materials and Methods

To investigate the risks of radiation exposure on pregnant women and fetuses, this paper reviewed the literature through basic investigations in radiology. The electronic databases used were PubMed, EBSCO, and RSNA. Search terms such as ‘pregnant woman’, ‘fetus’, ‘embryo’, and ‘pregnancy’ were utilized, including modifiers that may have included words regardless of the suffix. The literature review began in January 2017 and ended in September 2021. Full articles of related studies were searched, and additional related studies not identified in the initial search were identified through searches of books. Afterwards, inclusion criteria were applied to potentially relevant studies, and study data were extracted and tabulated.

From a technical point of view, the research was conducted as a literature review, and the topic of interest was determined by synthesizing previously published papers and books. The investigation was conducted in two major stages. First, through a database search, papers related to the purpose of this study were read, and papers to be used in this study were selected. Second, by citing books to support the rationale, radiation risks according to the period of pregnancy were investigated, and the best conditions for minimizing radiation effects on embryos and fetuses were sought. From the collected data, radiation sensitivity and radiation resistance according to the development of the fetus were analyzed.

## 3. Results

High radiation exposure can cause harmful effects on the human body. When pregnant women are exposed to radiation, the embryo or fetus will also retain some of the radiation. When the human body is exposed to low doses of radiation for a longer period than natural, biological effects occur, which are also known as stochastic effects and are considered long-term risks [22]. Radiation effects vary depending on the location and the characteristics of the local tissue and organ, dose level, period of exposure, and related disease. Humans can expect to reduce their lifespan by approximately 10 days for every 10 mSv [23]. Radiation-induced malignancies, shortened lifespans, and genetic damage are common biological effects of radiation [24,25]. Radiation-induced malignancies include leukemia, skin carcinoma, thyroid cancer, breast cancer, osteosarcoma, and lung cancer [23]. In contrast, deterministic effects, also known as non-stochastic effects, are considered short-term risks [22]. Deterministic effects occur during serious radiation accidents and do not occur during routine diagnostic imaging examinations or routine occupational exposure [26,27]. These effects increase when radiation exposure doses increase. Examples of deterministic effects include skin erythema, cataracts, sterility, fibrosis, hematopoietic damage, and fetal death. As the cells die under the influence of radiation, it causes obvious dysfunction in tissues or organs, and biological effects will occur at this time [28].

Acute radiation exposure can also cause a group of symptoms known as a syndrome. The lethal dose, which can kill more than 50% of the population in 30 days, is between 2500–4500 mSv [27]. When a person is exposed to acute radiation, they can develop hematologic syndrome; gastrointestinal syndrome; central nervous syndrome; or local tissue damage to the skin, eye, and gonads. The survival time of a person exposed to acute radiation ranges from 15 min to 50 days depending on the dose; when exposure doses are increased, survival times and the number of survivors decreased [27].

Embryos and fetuses are more sensitive to radiation than adults because they have a rapidly developing cell system. During rapid cell proliferation, migration, and differentiation, the embryo suffers from the adverse effects of radiation because it is developing under a dynamic system [27]. The effects of radiation on fertility also vary depending on the gestational age of the pregnancy. The first trimester during pregnancy is the most radiosensitive period [23]. To measure the approximate fetal dose, a thermoluminescent dosimeter was used [29]. When pregnant women were exposed to primary radiation, their fetus was exposed to about one-third of the entrance radiation dose [29]. The fetus receives less radiation than the mother, but continuous radiation exposure can cause harmful effects.

Many institutions have released policies regarding the harmful effects radiation can have on the fetus. The National Council on Radiation Protection and Measurement (NCRP) issued a report stating that when pregnant women are exposed to less than 50 mSv it is considered a negligible risk for an abnormality, and when exposed to more than 150 mSv it is considered a risk for malformation [30,31]. This guideline is important because healthcare providers have to inform pregnant women about the benefits and risks of radiation. The International Commission on Radiological Protection (ICRP) publicized the radiological protection and guidelines procedures for pregnant women, stating that at 100 mSv and above the determination of radiation exposure information for the embryo or fetus should be based on the expected dose for and the risk of harm to the developing embryo or fetus [25,32]. Having information on estimated doses, the benefits of diagnostic imaging, and the risks of radiation exposure helps pregnant women make informed decisions.

Because biological effects last a lifetime, the ICRP also released a policy limiting exposure doses [25,32]. The policy states that physicians and healthcare providers must take reasonable steps to avoid unnecessary sources of exposure and reduce exposure doses when imaging is necessary [25,32]. Appropriate policies such as these help physicians and healthcare providers think of reasonable steps to avoid unnecessary radiation exposure.

## 4. Discussion

To minimize the harmful effects of radiation on the fetus, all parties must understand the degree of radiation risk at the various stages of pregnancy. Radiation exposure between 8–15 weeks of gestation has the highest risk for fetal abnormalities [11]. As previously stated, fetal death is a deterministic effect. Embryos are relatively radiation-tolerant in the preimplantation stage but are very sensitive to radiation during the organ-forming (weeks 2–8) and neural stem cell proliferation stages (weeks 8–15) [28]. When the embryo enters the organogenesis and neuronal stem cell proliferation phases, the rapidly developing cell system begins to form and enters the most radiosensitive phase. Many researchers have found that after 25 weeks evidence-based medicine does not support an increased risk of effects associated with higher radiation doses [8,11,28]. Depending on the radiation dose and fetal development phase, effects such as embryonic death, malformation, growth restriction, and miscarriage vary greatly. The following chart outlines the fetal effects from low-level radiation exposure in detail by using both animal and human studies; the collected data represent the most radiosensitive phase and most radioresistant phase during fetal development (Table 1).

The following are some risks of medical X-ray imaging on fetal development based on radiation dose and fetal development phase [2,34,35,36,37,38]:During the preimplantation phase of embryonic development, radiation exposure is fatal to the embryo. Beyond this phase, there is no risk from radiation exposure below 100 mSv;Exposure less than 2 weeks post-conception up to 10 rads (100 mSv) may lead to embryonic death;Exposure at 2–7 weeks post-conception up to 5–50 rads (50–500 mSv) may lead to increases in major malformations as well as growth restriction. Exposure greater than 50 rads may lead to a substantial risk of malformation, growth restriction, and miscarriage;Exposure during major organ-forming periods, usually between 3–8 weeks, can result in abnormalities in the organs, with a threshold of 100 mSv;Between 8–25 weeks, the central nervous system shows high sensitivity to radiation, and severe intelligence degradation is at a high probability;Beyond 25 weeks, the accumulation of nitrogen, body fat, calcium, water, mineral, and nitrogen in the fetus increases; therefore, fetal sensitivity to radiation is small.

To find the effective dosage range of an embryo, the radiation effects on laboratory mice at gestational day 10 were analyzed. This study found that 0.5 Sv and less did not affect mice embryonic development, 0.5–1 Sv was mildly effective, 1–2 Sv was severely effective, and 2–4 Sv was lethal [39,40]. Furthermore, exposure greater than 5 Sv led to retardation of development, aberrant growth, malformation, or intrauterine mortality [37,38,39,40].

According to the fetal impact report, the biological effects that fetuses can experience delayed growth, organ deformities, small head size, severe mental retardation, reduced IQ, and childhood cancer [8,41]. If radiation dose exposure occurs during week 1 of fetal implantation, the fetus may die. In one example, the children of atomic bomb survivors from Hiroshima and Nagasaki who were exposed prenatally to more than 0.2 Sv were found to be 2–3 cm smaller and 3 kg lighter and have a head circumference 1 cm smaller than the control group. About 25% of the children who had a smaller head size were mentally retarded [8,41]. A radiation exposure dose less than 0.1 Sv during the stem cell proliferation phase at 8–15 weeks had almost no effect, and a decrease of 0.025 in IQ per Sv was seen over 0.1 Sv. At 11 weeks, the most likely radiation effect to occur was childhood cancer, with leukemia being the most common type [42].

The Centers for Disease Control and Prevention outlined the estimated risks of a person developing cancer based on prenatal radiation exposure [43]. Table 2 represents the lifetime risk of cancer in children due to prenatal radiation exposure as published by the ICRP [9,25,43]. Lifetime cancer mortality is about one-third of the number of cases, but childhood cancer mortality is only one-half of the number of cases. The death rate for childhood cancer is higher than the existing mortality rate compared to the number of cases.

Children have an approximately 38% chance of developing childhood cancer without radiation exposure except the background radiation. Exposure to 0.00–0.05 Sv of radiation can cause childhood cancer with a 0.3% chance; this percentage is similar to that of not being exposed (Table 2). If exposed to 0.05–0.5 Sv, children have a 0.3–1% chance of developing childhood cancer. If exposed to 0.5 Sv or more, children have a 1–6% chance of developing childhood cancer [43,44]. Many studies have noted that radiation exposure during pregnancy increases the chances of developing cancer in the born child by more than 6% per Sv [45,46,47,48,49].

The epidemiologic data on the atomic bomb survivors revealed that 0.1–0.2 Sv could affect fetal injury during the gestational period [50]. Radiation exposure during pregnancy could cause excess fetal loss and brain injury, microcephaly, and mental retardation. Even those who were up to 6500 feet away from the center of the explosion experienced miscarriages [50,51,52,53,54]. Atypically small heads were observed among some of the children exposed to atomic radiation during 3–17 weeks of gestation [8,55].

Statistics of pregnant women who were exposed to radiation over a decade show that fetuses are exposed to an average of 0.43 mSv per general radiography, 3.24 mSv per general radiography with primary radiation, 4.3 mSv per CT scan, and 2.91 mSv per fluoroscopic scan [55]. Fetal radiation exposure has increased 2.6 times from 2.1 mSv in 2006 to 0.82 mSv in 1997 [55]. As the frequency of radiography increases steadily, so too does the radiation exposure to pregnant women and fetuses, and the best decision should be made in consideration of the impact on the fetus [1,55].

However, for pregnant trauma or emergency patients, radiography, especially a CT procedure, might be necessary. Patients are exposed to an average of 4.3 mSv during chest CT scans and over 30 mSv during CT angiograms [55,56]. Thus, an appropriate non-ionizing testing method, such as a magnetic resistance imaging scan or an ultrasound, should be used to reduce the probability of cancer or mental retardation in the fetus [57].

### 4.1. Dose Limit to Pregnant Women

According to the NCRP, if the cumulative radiation dose is less than 0.05 Sv and the background radiation dose is less than 0.001 Sv during month 9 of gestation, the fetus will not be affected [33]. The Radiation Safety Committee of the U.S. Center for Death Control and Prevention recommends that pregnant radiologists and radiologic technologists avoid being exposed to more than 0.005 Sv of a cumulative radiation dose during pregnancy [33,43,58,59]. Occupational radiation exposure is typically measured with a dosimeter badge, and these devices are not required for pregnant patients performing diagnostic radiography [43,58,59,60,61].

### 4.2. Ethics in Radiology

The four biomedical ethical principles are autonomy, beneficence, non-maleficence, and justice [62,63]. Autonomy refers to the wishes of a patient according to their self-rule to protect themselves from harm. Healthcare providers must receive informed consent to obtain patient information, and patients can make decisions voluntarily depending on their preferences. Beneficence and non-maleficence are related to each other. The Hippocratic imperative is explained by beneficence (bring benefits) and non-maleficence (do no harm). Healthcare providers must bring benefits to their patients by following their patients’ interests and not causing harm. Healthcare providers must also treat their patients equally to ensure justice [62,63,64,65,66].

ICRP Task Group 94 on the ethics of radiological protection and many researches review the medical ethics and ethical values in radiation protections of patients [63,67]. The ICRP is specialized in radiation protection based on the scientific evidence background to support the medicine, research, and education [9,25,32,63,67]. These principles help healthcare providers make better decisions for patients. There are times when ethical decisions have no right or wrong answer and only better decisions can be made in a specific situation. Healthcare providers should consider beneficence and the harmful effects that radiation can have on pregnant women to minimize harm and provide benefits. It is an important virtue of healthcare providers to make better decisions for patients through communication [57]. Communicating with patients maximizes the likelihood of achieving good results and patient-centered outcomes [68]. Providing an assessment of the risks and benefits of a procedure is necessary to apply the principles of beneficence and justice. When procedures have both harmful and beneficial outcomes to patients, they must bring significant benefit with a small risk of harm according to the ethical principles [62]. Following these principles provides guidelines for beneficial outcomes.

When pregnant women and the fetus is exposed from radiation for the diagnostic purpose, physicians must think about what ethics to follow in that particular situation. If it is not a life-threatening situation of pregnant women, radiation exposure may not provide significant benefits as opposed to harm. A single diagnostic imaging procedure may not cause a risk for injury to pregnant women and the fetus; however, long-term use of radiation can have harmful effects. Annals of the ICRP state that if a pregnant patient undergoes a radiographic examination justification for the examination should be evaluated. Then, the procedure must be optimized with strict adherence to the best techniques to proceed with the inspection [32]. When physicians and radiologists make decisions regarding X-ray procedures, they follow these principles to judge between the benefits and risks. Briefing the patient and, ultimately, obtaining their consent is necessary to thoroughly communicate the risks and benefits of radiation [44]. Further, physicians should continuously evaluate their decisions to ensure justification [69].

Even when they decide to proceed with an X-ray, pregnant women may still have concerns about the effects that radiation can have on their fetus. Surveys have demonstrated that less than 10% of healthcare providers know the annual radiation limit dose for pregnant women, 22% of emergency department physicians offer the risks and benefits of radiography examination to patients, and 7% of patients note having received information [12,18]. Radiologic technologists, physicians, and radiologists have to be educated on the various effects that radiation can have on pregnant women in order to provide them with accurate information. Proper explanations and more communication will help reduce pregnant women’s concerns. In this manner, patients can make informed decisions based on good communication with their healthcare provider.

### 4.3. Ethical Dilemma in Radiology

The ethical dilemma between radiologic technologists and healthcare providers is increasing due to a lack of awareness of radiation exposure risks. Recent studies have shown that some healthcare providers have little knowledge about dose limits and radiation risks associated with radiographic imaging of pregnant women [70]. Recent studies have also shown that neither emergency medical doctors nor patients are aware of the risk of cancer caused by medical radiation, such as CT [71]. Little knowledge about radiation exposure risks can cause severe health risks for patients, especially pregnant women and fetuses.

Ethics are necessary to make appropriate decisions for better treatment. Ethics require a rigorous process of collecting relevant information to formulate an argument, such as medical decision-making [72]. In other words, a reasonable and coherent argument is essential in medical ethics [73]. Arguments and disagreements can help physicians and radiologists make better decisions because different perspectives and opinions are being shared. In making decisions, deep conversations with those involved are desirable and essential for obtaining relevant information about the patient [74]. To avoid miscommunication and mistrust, physicians must obtain consent from pregnant patients before performing diagnostic imaging procedures.

## 5. Conclusions and Future Study

The highest exposure of radiation to pregnant women poses the greatest risk for fetal abnormalities. Prenatal radiation exposure can also cause cancerous diseases as well as non-cancerous diseases. The purpose of this literature review was to investigate the significance of the relationship between radiation exposure on pregnant women and resulting diagnostic issues. Educating healthcare providers about the risks that radiation can have on pregnant women is important for minimizing unnecessary radiation exposure risks.

Quantitative methods are necessary to answer the given research questions and obtain answers from healthcare providers. To assess the perspectives of healthcare providers, a 20-question survey will be collected from radiologic technologists, physicians who are working in a labor and delivery department, and radiologists who are working at a children’s hospital. The survey will be anonymous and ask some questions about their understanding of radiation exposure, the harmful effects of radiation on pregnant women, the radiation risks to the fetus, and the benefits of radiation exposure depending on the disease. Once these surveys have been completed, the results will be analyzed to create an educational session for the radiology department, in which radiologic technologists, radiologists, and pediatric physicians will participate. This session will allow everyone in attendance to appreciate the harmful effects and benefits of radiation exposure on pregnant women and fetuses as well as minimize unnecessary radiation exposure by improving overall communication.

## Figures and Tables

**Table 1 bioengineering-09-00236-t001:** The relationship between fetal development phase and radiation effects [8,11,28,33,34,35,36].

Period	Fetal Development Phase	Radiation Effects
1–6 days Embryonic period	Preimplantation phase	(Radioresistant phase) Prenatal death
2–8 weeks Gestational period	Organogenesis phase	(Radiosensitive phase) Growth retardation Organ malformation Small head size
8–15 weeks Embryonic period	Stem cell proliferation phase	(Radiosensitive phase) Severe mental retardation Reduction of IQ Small head size Childhood cancer (11th week)
Beyond 25 weeks Second trimester (26th week) Third trimester (28th week)	Less sensitive phase	Radioresistant phase

**Table 2 bioengineering-09-00236-t002:** Estimated risk for cancer in children from prenatal radiation exposure [9,25,43,44,45,46,47,48,49].

Radiation Dose	Estimated Childhood Cancer Incidence	Lifetime Cancer Mortality
No radiation exposure above background	0.3%	38%
0.00–0.05 Sv (0–5 rads)	0.3–1%	38–40%
0.05–0.5 Sv (5–50 rads)	1–6%	40–55%
>0.5 Sv (50 rads)	>6%	>55%

## Data Availability

Not applicable.

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
