# Peer review of "Diagnostic Imaging of Pregnant Women and Fetuses: Literature Review"

_bioengineering, 2022, doi:10.3390/bioengineering9060236_

Round 1
Reviewer 1 Report
The authors reviewed the risk of radiation for pregnant women and fetuses and discussed ethical problems. Suggestions described in the conclusion part are significant for medical staff.
There is nothing particular to correct, but maybe the comments about the risk of contrast mediums should be needed. Most of the medical staff (except obstetricians) also have a question about these agents in enhanced radiography for pregnant women. Please provide a short comment with literature for the agents.
Author Response
Thanks for your suggestion. Yes, I researched this topic to suggest appropriate cognition between radiation and pregnant women for medical staffs. I understand your point of view. At this research I focused on between radiation dose and fetus & pregnant women. For the further study, I will definitely research on the effects of contrast agent on the pregnant women and fetus. I really appreciate your suggestion ?

Reviewer 2 Report
General Remarks:
Units: rads and Gy are used randomly, although converted to one another and with no justification. SI units should be written in first place and, if necessary, traditional units should appear between parenthesis.
The paper only mention dose (gray, Gy). However, que concepts of equivalent dose, effective dose or dose equivalente (Sievert, Sv), could be appropriate in some sections, since we are dealing in the context of radiation protection were these quantities are relevant. E.g. in line 224, where the occupational radiation exposure is clearly mentioned, the sievert could be used instead of gray as well as effective dose. Also, the “badge” (line-227) measures the operational quantity “equivalent dose”, as a surrogate of the “effective dose”. When placed on the abdomen, it is used as a surrogate of the uterus dose equivalent in pregnant women. No considerations are made in the text about these details.
Particular remarks:
Line 21 – radiation dose of whom? Professionals, members of the public? Reference needed.
Line 23 – “collective dose”, instead of “radiation volume”
Line 26 - Sugestion: Include Dosedatamed 2 results and remove ref. 3 (reference 3 does not even presents the name of the journal)
Line 46 – The sentence: “It states that the maximum radiation dose for a fetus is 100 mGy; a fetal dose over 100 mGy could be a primary reason for terminating a pregnancy [8]” should be revised with care. Reference [8] does not claim that above 100 mSv the pregnancy should be terminated. Although the text does not state that either, a misinterpretation can be made easily.
Line 81 – The risk to pregnant women is quantified as “double”. This is not accurate as should be removed or justified.
Line 175 – “Biological [radiation] effects are caused by a certain amount of radiation” seems redundant because it claims the obvious.
Line 176 – Should be used “experience” or “include”, but not both.
Line 195 – “without radiation exposure” is inaccurate since the authors refer to the background radiation exposure.
Line 203 – “Excessive radiation exposure during pregnancy” is a subjective statement and lack quantification.
Line 251-254 – This sentences present a lack of objectivity and clarity that appear in several other sections of the paper. One should not state that simply that “exposure is harmful to pregnant women and the fetus” since, being correct if taken “per se” and decontextualized, the act of exposing someone to ionizing radiation must be justified and so, presents always some degree of benefit. Being so, one can also say that “exposure is benefic to pregnant women and the fetus” given the above conditions. In the same line it is stated that “If it is not a life-threatening situation, radiation exposure may not provide significant benefits as opposed to harm”. Clearly, even in a non-life-threatening situation this can be also true, given that the benefit to the woman or the child surpasses the potential harm.
Line 277 – Why “dose measurements” instead of “dose values” and/or “dose limits”?
Section 4.2 and section 4.3 (line 229 to line 291) – The authors refer to biomedical principles of Beauchamp and Childress. However, in the context of radiation protection, it would be better to mention also the ethical core values of ICRP (Publication 128, 2018) and also the “pragmatic value set” by Malone and Zölzer, 2016 (doi:10.1259/bjr.20150713)
Line 297 – Although “education healthcare providers” is frequently mentioned in the text, no suggestions on how to do it globally, besides the proposed future work, are presented. An educational scheme could be discussed, and a comparison with the ones that are being applied, depending on the professional position, with some external references of already existing recommendations, would benefit the work.
Author Response
General Remarks:
→ Units: rads and Gy are used randomly, although converted to one another and with no justification. SI units should be written in first place and, if necessary, traditional units should appear between parenthesis.
The paper only mention dose (gray, Gy). However, que concepts of equivalent dose, effective dose or dose equivalente (Sievert, Sv), could be appropriate in some sections, since we are dealing in the context of radiation protection were these quantities are relevant. E.g. in line 224, where the occupational radiation exposure is clearly mentioned, the sievert could be used instead of gray as well as effective dose. Also, the “badge” (line-227) measures the operational quantity “equivalent dose”, as a surrogate of the “effective dose”. When placed on the abdomen, it is used as a surrogate of the uterus dose equivalent in pregnant women. No considerations are made in the text about these details.
I totally agree with you. When I set the unit in the beginning, I followed the unit of the guidelines, book, and references. I agree with you that I need to change all the unit to the effective dose which is Sievert. I changed all the unit into Sv from Gy.
Particular remarks:
→ Line 21 – radiation dose of whom? Professionals, members of the public? Reference needed.
Thanks for your suggestions, this radiation doses from medical procedures to publics. I added the reference to support this sentence.
“Thanks to technological improvements, overall radiation doses from medical pro-cedures has been increased in the past 25 years [1].”
→ Line 23 – “collective dose”, instead of “radiation volume”
Yes, I modified this sentence.
“The American Society of Radiologic Technologists reported in the Radiation Safety Compliance Journal that the overall collective dose has increased nearly sixfold since the early 1980s.”
→ Line 26 - Sugestion: Include Dosedatamed 2 results and remove ref. 3 (reference 3 does not even presents the name of the journal)
Thanks for the suggestion, I deleted reference 3.
→ Line 46 – The sentence: “It states that the maximum radiation dose for a fetus is 100 mGy; a fetal dose over 100 mGy could be a primary reason for terminating a pregnancy [8]” should be revised with care. Reference [8] does not claim that above 100 mSv the pregnancy should be terminated. Although the text does not state that either, a misinterpretation can be made easily.
Thanks for your suggestion, I deleted the sentence to avoid the misinterpretation.
→ Line 81 – The risk to pregnant women is quantified as “double”. This is not accurate as should be removed or justified.
I agree with you. I removed that part to make accurate sentence.
“When pregnant women are exposed to radiation, the embryo or fetus will also retain some of the radiation.”
→ Line 175 – “Biological [radiation] effects are caused by a certain amount of radiation” seems redundant because it claims the obvious.
Thanks for your suggestion, I deleted that part to make clearer!
→ Line 176 – Should be used “experience” or “include”, but not both.
I agree with you. I deleted include in the sentence!
→ Line 195 – “without radiation exposure” is inaccurate since the authors refer to the background radiation exposure.
Thanks for your suggestion, I added “except the background radiation” to support the sentence.
“Children have an approximately 38% chance of developing childhood cancer without radiation exposure except the background radiation.”
→ Line 203 – “Excessive radiation exposure during pregnancy” is a subjective statement and lack quantification.
Thanks for the suggestion, I deleted “excessive” since it is subjective statement.
“Radiation exposure during pregnancy could cause excess fetal loss and brain injury, microcephaly, and mental retardation.”
→ Line 251-254 – This sentences present a lack of objectivity and clarity that appear in several other sections of the paper. One should not state that simply that “exposure is harmful to pregnant women and the fetus” since, being correct if taken “per se” and decontextualized, the act of exposing someone to ionizing radiation must be justified and so, presents always some degree of benefit. Being so, one can also say that “exposure is benefic to pregnant women and the fetus” given the above conditions. In the same line it is stated that “If it is not a life-threatening situation, radiation exposure may not provide significant benefits as opposed to harm”. Clearly, even in a non-life-threatening situation this can be also true, given that the benefit to the woman or the child surpasses the potential harm.
I understand your point of view. I agree with you that this statement could be misleading sentence and decontextualized. I modified that sentence as following.
“When expose the radiation to pregnant women and the fetus for the diagnostic purpose, physicians must think about what ethics to follow in that particular situation. If it is not a life-threatening situation of pregnant women, radiation exposure may not provide significant benefits as opposed to harm.”
→ Line 277 – Why “dose measurements” instead of “dose values” and/or “dose limits”?
Thanks for your suggestion. I modified to “dose limits”!
→ Section 4.2 and section 4.3 (line 229 to line 291) – The authors refer to biomedical principles of Beauchamp and Childress. However, in the context of radiation protection, it would be better to mention also the ethical core values of ICRP (Publication 128, 2018) and also the “pragmatic value set” by Malone and Zölzer, 2016 (doi:10.1259/bjr.20150713)
Thanks for the suggestion. I understand the point of your view. I added ICRP report and “pragmatic value set” manuscript to support my idea of ethical core values.
“ICRP Task Group 94 on ethics of radiological protection and many researches review the medical ethics and ethical values in radiation protections of patients [63, 67]. The ICRP is specialized in radiation protection based on the scientific evidence background to support the medicine, research, and education [9, 25, 32, 63, 67].”
→ Line 297 – Although “education healthcare providers” is frequently mentioned in the text, no suggestions on how to do it globally, besides the proposed future work, are presented. An educational scheme could be discussed, and a comparison with the ones that are being applied, depending on the professional position, with some external references of already existing recommendations, would benefit the work.
Thanks for your suggestion. When I worked at the Loma Linda Unviersity Children’s Hospital, I planned to have the survey of the healthcare providers to provide the educational forum. That is why I included in this further study plan to perform the survey and do the education session for the medical staffs in the hospital.
Round 2
Reviewer 2 Report
The suggestions have been properly addressed.